# An agent-based model to simulate the transmission dynamics of bloodborne pathogens within hospitals

Paul Henriot[1,2,3]*, Mohamed El-Kassas[4], Wagida Anwar[5], Samia A. Girgis[6], Maha El Gaafary[5], Kévin Jean[1,2,7], Laura Temime[1,2]

**1** Laboratoire Modélisation, Épidémiologie Et Surveillance Des Risques Sanitaires, Conservatoire national des arts et métiers (CNAM), Paris, France, **2** Unité PACRI, Risques Infectieux Et Émergents, CNAM-Institut Pasteur, Paris, France, **3** UMR EPIA, INRAE, Marcy-l'étoile, France, **4** Endemic Medicine Department, Faculty of Medicine, Helwan University, Cairo, Egypt, **5** Department of Community, Environmental and Occupational Medicine, Faculty of Medicine, Ain Shams University, Cairo, Egypt, **6** Department of Clinical Pathology, Faculty of Medicine, Ain Shams University, Cairo, Egypt, **7** IBENS, Ecole normale supérieure, CNRS, INSERM, Université Paris Science & Lettres, Paris, France

\* paul.henriot@protonmail.com

## Abstract

Mathematical models are powerful tools to analyze pathogen spread and assess control strategies in healthcare settings. Nevertheless, available models focus on nosocomial transmission through direct contact or aerosols rather than through blood, even though bloodborne pathogens remain a significant source of iatrogenic infectious risk. Herein, we propose an agent-based SEI (Susceptible-Exposed-Infected) model to reproduce the transmission of bloodborne pathogens dynamically within hospitals. This model simulates the dynamics of patients between hospital wards, from admission to discharge, as well as the dynamics of the devices used during at-risk invasive procedures, considering that patient contamination occurs after exposure to a contaminated device. We first illustrate the use of this model through a case study on hepatitis C virus (HCV) in Egypt. Model parameters, such as HCV upon-admission prevalence and transition probabilities between wards or ward-specific probabilities of undergoing different invasive procedures, are informed with data collected in Ain Shams University Hospital in Cairo. Our results suggest a low risk of HCV acquisition for patients hospitalized in this university hospital. However, we show that in a low-resource hospital, frequent device shortages could lead to increased risk. We also find that systematically screening patients in a few selected high-risk wards could significantly reduce this risk. We then further explore potential model applications through a second illustrative case study based on HBV nosocomial transmission in Ethiopia. In the future, this model could be used to predict the potential burden of emerging bloodborne pathogens and help implement effective control strategies in various hospital contexts.

## Author summary

Bloodborne pathogens (BBPs) such as HCV, HBV or HIV, are a major public health concern as they can lead to a variety of medical conditions, including cirrhosis and

**Data availability statement:** The data analysed in this study is available upon request only. De-identified data cannot be publicly shared, as our study involves sensitive data on human participants, and could be indirectly identifying based on multiple patient characteristics. Individual data requests may be sent to the CorC (secr-CORC@pasteur.fr). The code of the model is available in a GitHub repository: https://github.com/phenriot/BloodPaTH

**Funding:** PH was funded by Agence Nationale de Recherches sur le Sida et les Hépatites Virales (ANRS), Grant Number 12320 B115. The funder had no role in study design, data collection and analysis, decision to publish, or preparation of the manuscript.

**Competing interests:** The authors have declared that no competing interests exist.

cancers with significant mortality and morbidity. If infection control measures are inadequate, transmission of such pathogens to patients can occur via contaminated devices during invasive procedures within healthcare settings. The aim of our work was to build a tool to help assess the potential for BBP transmission to patients using mathematical modelling. Using data collected in an Egyptian hospital we evaluated the risk of HCV infection for hospitalized patients in two different scenarios: the case of a high-resource setting and the one of a low-resource setting. We found that although the risk of infection in high-resource settings is low, frequent device shortages could lead to increased risk in low-resource settings. We also found that screening patients upon admission in the setting or in specific wards could lead to a significant risk reduction. Our model was further tested on nosocomial HBV transmission in an Ethiopian hospital. In the future, this model could be used to predict the potential burden of emerging BBPs.

## 1. Introduction

Bloodborne pathogens are a major public health concern as they can lead to a variety of medical conditions, including cirrhosis and cancers with significant mortality and morbidity [1]. Three viruses are of major concern: hepatitis B and C viruses (HBV and HCV) and human immunodeficiency virus (HIV). Injecting drug use is considered as one of the main routes of bloodborne pathogen diffusion in the community, but iatrogenic transmission may occur during invasive procedures when compliance with infection control measures is imperfect [2]. Recent studies have highlighted the persisting increased risk of HBV and HCV infection in individuals exposed to invasive procedures, with high-risk levels in some low- or middle-income countries [3,4].

Mathematical models are widely used to better understand pathogen transmission and assess control strategies, particularly within healthcare settings [5]. Nevertheless, models investigating the bloodborne transmission of pathogens remain rare and often focused on community-level epidemics, notably among drug users [6]. No dynamic model has been proposed to specifically explore within-hospital transmission routes of bloodborne viruses and evaluate control measures.

Herein, we present a flexible agent-based model describing movements (from admission to discharge) of patients between wards in a hospital, their exposure to pathogen-contaminated blood through medical devices during invasive procedures, and the ensuing epidemic process. We illustrate the use of this model through case studies on HCV in Egypt, relying on detailed patient observation data collected in a Cairo hospital and HBV in Ethiopia. We explore the impact of device availability within hospitals and assess the effect of different screening interventions on the risk of infection for hospitalized patients.

## 2. Methods

We developed a stochastic, discrete-time, individual-based model that describes the dynamics of patients within a hospital, from admission to discharge, as well as the dynamics of medical and surgical devices used during at-risk invasive procedures. The model further simulates the between-patient spreading dynamics of a bloodborne pathogen through contaminated devices (Fig 1). Transmission occurs during invasive procedures: device contamination from an infectious patient to an uncontaminated device and patient infection from a contaminated device to a susceptible patient. Herein, we describe the mathematical framework

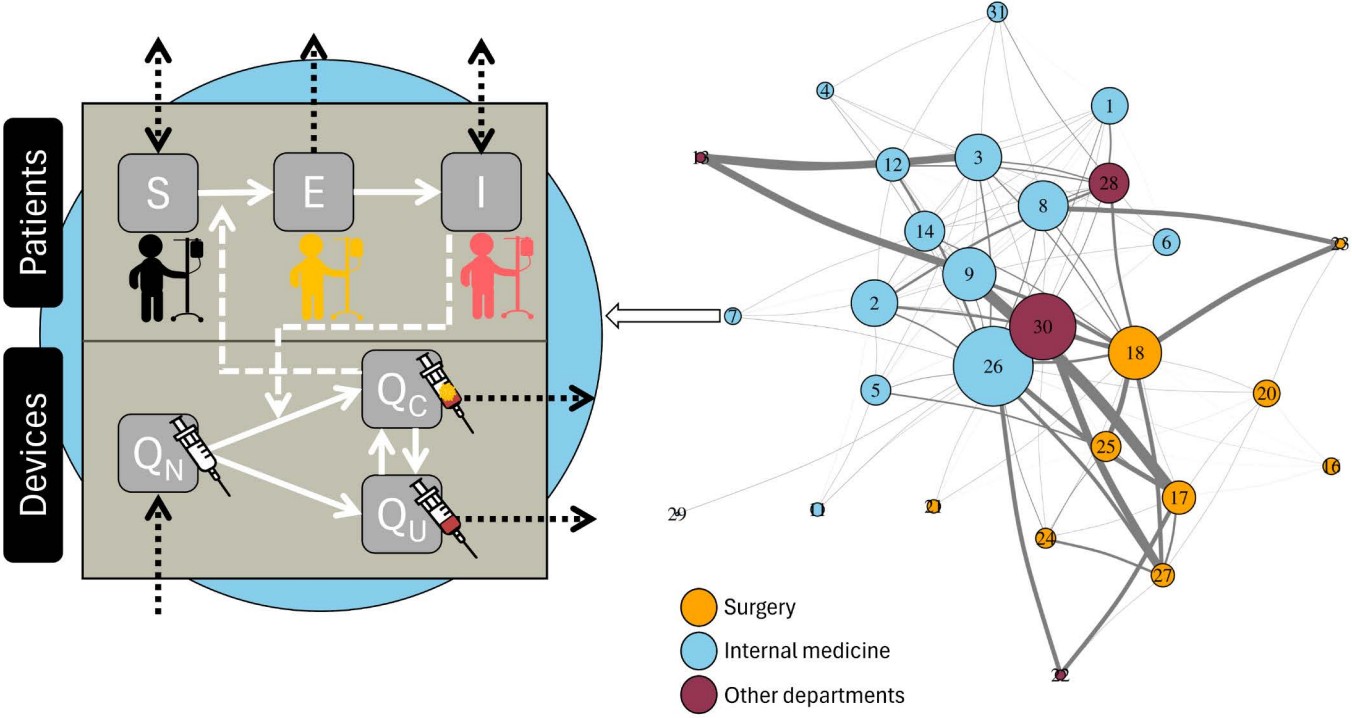

**Fig 1. Model framework.** Patients are categorized as either Susceptible (S), Exposed (E), or Infectious (I), while the available device may either be new ($Q_N$) and sterile, previously used but uncontaminated ($Q_U$), or previously used and contaminated ($Q_C$). The ward network reflects the structure of Ain Shams hospital and is used as an illustrative example to apply our model. Hospital wards are depicted as blue (for internal medicine wards), yellow (for surgery wards), or purple (for other wards visited by internal medicine and surgery patients) circles, with circle size proportional to the number of beds in the ward. The width of between-ward edges is proportional to the corresponding number of patient movements. The full list of ward names associated with each ward number is provided in S1 Table.

of the model as well as its specificities from a user point of view, its parametrization, and operation.

## 2.1. Population dynamics

Tables 1 and 2 summarise the model's parameters. The total hospitalized population is assumed to remain constant over time at hospital capacity. $N_{pat}$ , so that the number of patients leaving the hospital at time $t$ is equal to the number of patients entering the hospital at time $t+1$. Patients may belong to any of $N_g$ distinct profiles such as age, department of admission, etc... For each profile $g$ and each ward $w$, the upon-admission prevalence is defined as $Prev_w^g$ .

At each time step, a patient with profile $g$ can move from ward $i$ to ward $j$ with probability $\tau_{i,j}^g$ . All transfer probabilities are provided in the square transfer matrix $M_{tr}^g$ of size $(N_w+1)\times(N_w+1)$ with the last row being a vector of zeros except at the last position (i.e., every discharged patient stays in the "discharged" compartment):

$$M_{tr}^g = \begin{bmatrix} \tau_{1,1}^g & \cdots & \tau_{1,j}^g \\ \vdots & \ddots & \vdots \\ \tau_{i-1,1}^g & \cdots & \tau_{i-1,j}^g \\ 0 & \cdots & 1 \end{bmatrix}$$

**Table 1. Hospital-associated parameters.** "Data" refers to data collected in Ain Shams hospital. More details are provided in S3–S10 Tables.

| Entry parameters | Notation | Value | | Reference |
|---|---|---|---|---|
| Initial quantity of new device $k$ in ward $w$ | $q_{n,k,w}^{init}$ | S7–S10 Tables | | |
| Initial quantity of previously used uncontaminated device $k$ in ward $w$ | $q_{u,k,w}^{init}$ | | | Data |
| Initial quantity of previously used contaminated device $k$ in ward $w$ | $q_{c,k,w}^{init}$ | | | |
| Probability of successful sterilization for device $k$ | $P_{ster,k}$ | 0.95 (for all devices in the high-resource setting) 0.80 (for all devices in the low-resource setting) | | Expert opinion |
| Resupply frequency for device $k$ | $f_k$ | S1 Text | | Estimated from data |
| | | **Surgery** | **Internal medicine** | |
| Total number of patients (capacity) | $N_{pat}$ | 400 | 570 | Data |
| Number of patient profiles | $N_g$ | 1 | | Data |
| Probability of belonging to profile $g$ | $P^g$ | 0.41 | 0.59 | Data |
| Number of wards | $N_w$ | 11 | 15 | Data |
| Number of distinct procedures | $N_{proc}$ | 15 | | Data |
| Number of device types | $N_k$ | 10 | | Data |
| Profile-specific probability of undergoing procedure $p$ in ward $i$ | $\alpha_{i,p}^g$ | S5 Table | S6 Table | Data |
| Profile-specific transfer rate between wards $i$ and $j$ | $\tau_{i,j}^g$ | S3 Table | S4 Table | Data |

**Table 2. Pathogen-associated parameters.**

| Entry parameters | Notation | Value | Reference |
|---|---|---|---|
| Profile-specific upon-admission prevalence in ward $w$ | $Prev_w^g$ | S1 Table | Estimated from data |
| Transmission risk for procedure $p$ | $r_p$ | S4 Fig | Henriot et al., 2022 [4] |
| Duration of eclipse phase (min-max) | $(d_{e,min}, d_{e,max})$ | (2–14 days) (HCV) (15–15 days) (HBV) | Martinello et al., 2018 [9] Candotti et Laperche 2018 [12] |

## 2.2. Procedures

In total, $N_{proc}$ distinct invasive procedures are performed within the hospital. At each time step, a patient of group $g$ from ward $i$ undergoes procedure $p$ with probability $\alpha_{i,p}^g$. The procedure probability matrix of size $N_w \times (N_{proc} + 1)$, is denoted $M_{proc}^g$. The last column of this matrix is the vector of probabilities associated with the event "no procedure" for each ward (i.e., the patient stays in a given ward without undergoing any procedure):

$$M_{proc}^g = \begin{bmatrix} \alpha_{1,1}^g & \cdots & \alpha_{1,N_{proc}}^g & \left(1 - \sum_{N_{proc}}^{i=1} \alpha_{1,i}^g\right) \\ \vdots & \ddots & \vdots & \vdots \\ \alpha_{N_w,1}^g & \cdots & \alpha_{N_w,N_{proc}}^g & \left(1 - \sum_{N_{proc}}^{i=1} \alpha_{N_w,i}^g\right) \end{bmatrix}$$

Note that each row of this matrix sums to 1 and that a patient can only undergo one procedure at each time step.

## 2.3. Device dynamics

Devices are classified into $N_k$ types and associated with procedures by a table giving the device types in the first column and the associated procedures in the other columns. A procedure can be associated with the use of multiple devices.

The quantity of available devices is ward-dependent, type-dependent, and time-dependent. It is divided into three device groups: (a) new (sterile) device, (b) previously used uncontaminated device, and (c) previously used contaminated device.

The number of new available devices at time $t$ of type $j$ in ward $i$ is given by $q_n^{i,j}(t)$; quantities of new devices over the entire hospital are provided in a $\mathrm{Nw} \times \mathrm{N}_K$ sized matrix:

$$Q_n(t) = \begin{pmatrix} q_n^{1,1}(t) & \cdots & q_n^{1,k}(t) \\ \vdots & \ddots & \vdots \\ q_n^{w,1}(t) & \cdots & q_n^{w,k}(t) \end{pmatrix}$$

The number of previously used uncontaminated devices at time $t$ of type $j$ in ward $i$ is given by $q_u^{i,j}(t)$; quantities of used uncontaminated devices over the entire hospital are provided in a $Nw \times N_K$ sized matrix:

$$Q_u(t) = \begin{pmatrix} q_u^{1,1}(t) & \cdots & q_u^{1,k}(t) \\ \vdots & \ddots & \vdots \\ q_u^{w,1}(t) & \cdots & q_u^{w,k}(t) \end{pmatrix}$$

Finally, the number of previously used and contaminated devices at time $t$ of type $j$ in ward $i$ is given by $q_c^{i,j}(t)$; quantities of contaminated devices over the entire hospital are provided in a $\mathrm{Nw} \times \mathrm{N}_K$ sized matrix:

$$Q_c(t) = \begin{pmatrix} q_c^{1,1}(t) & \cdots & q_c^{1,k}(t) \\ \vdots & \ddots & \vdots \\ q_c^{w,1}(t) & \cdots & q_c^{w,k}(t) \end{pmatrix}$$

## 2.4. Transmission dynamics

**2.4.1. Patients.** Upon admission, a patient can be either Susceptible (S) or Infectious (I). During hospitalization, a Susceptible patient can become Exposed (E) after exposure to a reused contaminated material. We denote $\rho^s(t)$ as the status (i.e., 0 for S, 1 for E, or 2 for I) of patient $s$ at time $t$. Initial statuses of patients are randomly drawn according to a vector of upon-admission prevalences giving the initial prevalence (i.e., probability of being pathogen-positive when entering the hospital) for each group of patients.

A device of type $k$ is reused on patient $s$ at time $t$ for procedure $p$ in ward $w$ if no new device is available in the ward (i.e., $q_n^{k,w}(t) = 0$). The contamination status of the device is defined by $X_c^{k,w}$, which follows a Bernoulli law with $P\left(X_c^{k,w} = 1; t\right) = \dfrac{q_c^{k,w}(t)}{q_u^{k,w}(t) + q_c^{k,w}(t)}$ (contaminated device) and $P\left(X_c^{k,w} = 0; t\right) = 1 - \dfrac{q_c^{k,w}(t)}{q_u^{k,w}(t) + q_c^{k,w}(t)}$ (uncontaminated device).

The time-dependent probability of a susceptible patient getting infected after exposure to reused contaminated devices during procedure $p$ in ward $w$ is computed as:

$$P_{S \to E}(t) = 1 - \prod^{k \in \overline{v_p}} \left(1 - r_p \times P\left(X_c^{k,w}; t\right)\right)$$

Where $\vec{v}_p$ is the vector of device types used during procedure $p$, and $r_p$ is the risk of infection after exposure to a contaminated device during the same procedure, randomly drawn in an associated risk distribution.

If infection occurs, the patient becomes exposed, and his status changes to $\rho^s(t)=1$. The patient stays in the exposed state until the end of an eclipse phase $e$ is reached. This corresponds to the pre-ramp-up phase of the pathogen's natural history, during which the patient's infectiousness is considered null. The eclipse phase duration is drawn in a uniform distribution with parameters $e_{min}$ and $e_{max}$ for each new patient $s$ entering the hospital:

$$e^s \sim U(e_{min}, e_{max})$$

A counter $h^s(t)$ of the number of time-steps since exposure is initialized upon admission and is given at each time-step by:

$$\{ \begin{matrix} h^s(t+1)=0 \ if \ \rho^s(t)=0 \ or \ \rho^s(t)=2 \\ h^s(t+1)= \ h^s(t)+1 \ if \ \rho^s(t)=1 \end{matrix}$$

If $h^s(t) \geq e^{\ s}$, the eclipse phase is over, and the exposed patient becomes infectious ($\rho^s(t)=2$) until he leaves the hospital.

**2.4.2. Devices.** At each time step, in each ward, patients undergo their procedures successively. When $N_{pat}^{w,k}(t)$ patients in ward $w$ need to undergo a procedure requiring a device of type $k$, these procedures requiring the same device type are performed randomly. The rank of patient $s$ is then denoted $O_s^{w,k}(t)$ among all procedures requiring device $k$ at time $t$ in ward $w$.

Any device not already contaminated and used on an infected individual is considered contaminated after exposure. However, after each procedure, the device undergoes a sterilization process, which successfully clears contamination with probability $P_{ster}^k$.

To that aim, we define $Z_d^k$ as a device-dependent random variable following a Bernoulli distribution (i.e., the device is well disinfected or insufficiently disinfected) with $P(Z_d^k =1) = P_{ster}^k$ and $P(Z_d^k =0) = 1 - P_{ster}^k$.

Between times $t$ and $t+1$, the dynamics of device type $k$ in ward $w$ are thus described by the following systems of equation:

If $q_n^{k,w}(t+\varepsilon_{s-1})>0$:

$$\left[ \begin{matrix} q_n^{\ k,w}(t+\varepsilon_s)=q_n^{\ k,w}(t+\varepsilon_{s-1})-1 \\ q_u^{\ k,w}(t+\varepsilon_s)= \begin{cases} q_u^{\ k,w}(t+\varepsilon_{s-1})+1 \ if \ \rho^s(t) \in \{0,1\} \ or \ \rho^s(t)=2 \ and \ Z_d^k =1 \\ q_{nc}^{\ k,w}(t+\varepsilon_{s-1}) \ otherwise \end{cases} \\ q_c^{\ k,w}(t+\varepsilon_s)= \begin{cases} q_c^{\ k,w}(t+\varepsilon_{s-1})+1 \ if \ \rho^s(t)=2 \ and \ Z_d^k =0 \\ q_c^{\ k,w}(t+\varepsilon_{s-1}) \ otherwise \end{cases} \end{matrix} \right.$$

If $q_n^{k,w}(t+\varepsilon_{s-1})=0$:

$$\begin{cases} q_n^{k,w}\left(t+\varepsilon_s\right)=0 \\[2ex] q_{nc}^{k,w}\left(t+\varepsilon_s\right)=\begin{cases} q_u^{k,w}\left(t+\varepsilon_{s-1}\right)-1 \; if \; \left(X_c^{k,w}=0 \; and \; \rho^s(t)=2 \; and \; Z_d^{\,k}=0\right) \\ q_u^{k,w}\left(t+\varepsilon_{s-1}\right)+1 \; if \; \left(X_c^{k,w}=1 \; and \; Z_d^{k}=1\right) \\ q_u^{k,w}\left(t+\varepsilon_{s-1}\right) \; otherwise \end{cases} \\[4ex] q_c^{k,w}\left(t+\varepsilon_s\right)=\begin{cases} q_c^{k,w}\left(t+\varepsilon_{s-1}\right)+1 \; if \; \left(X_c^{k,w}=0 \; and \; \rho^s(t)=2 \; and \; Z_d^{\,k}=0\right) \\ q_c^{k,w}\left(t+\varepsilon_{s-1}\right)-1 \; if \; \left(X_c^{k,w}=1 \; and \; Z_d^{k}=1\right) \\ q_c^{k,w}\left(t+\varepsilon_{s-1}\right) \; otherwise \end{cases} \end{cases}$$

Where $\varepsilon_s = \dfrac{O_s^{w,k}(t)}{N_{pat}^{w,k}(t)}$ is the increment of time step generated by a procedure using device k performed on patient s and $N_{pat}^{w,k}(t)$ is the total number of patients undergoing a procedure requiring device $k$ in ward $w$ at time $t$.

Finally, each type of device undergoes a full renewal at a given device-dependent and ward-dependent frequency $f_k^w$. New devices then replace all previously used devices of that type.

## 2.6. Model application

### 2.6.1. Application to HCV in an Egyptian hospital: baseline scenarios.

We used detailed longitudinal data collected in an Egyptian hospital (Ain-Shams University Hospital, Cairo) in 2017 over six months to inform multiple parameters of our model. Five hundred patients were screened upon admission for HCV positivity and followed throughout their hospitalization in the Surgery and Internal Medicine departments. Many data were collected: (a) Upon-admission patient characteristics such as age, gender, and history of previous hospitalization, (b) patient location within the hospital, and (c) procedures undergone by these same patients. More details on the collected data are available in Anwar et al., 2021 [7].

These data were used to parametrize the model in the particular estimate: (a) The initial prevalence, (b) the number of patients, (c) the number of wards, (d) the number of procedures, (e) the transition matrix between wards within the hospital, and (f) the probability matrix associated with the procedures. These parameters were estimated separately for both departments (i.e., patient profiles). As the minimum duration of a procedure in our data was 5 minutes, we considered a 5-minute time-step for our transition matrix and procedures probability matrix estimations. In addition, we used other data collected for the year 2015 to inform the quantity of available supply within this same hospital, the date of resupply, as well as the admission probability in each department. The values of all these parameters and the method used to estimate the resupply date are detailed in the supplementary material (S3–S10 Tables and S1–S3 Figs and S1 Text).

Finally, per-procedure transmission risks were retrieved from a previously published risk-assessment [8], and the minimum and maximum durations of the eclipse phase were found in the literature [9]. No information was available to estimate the probability of successful sterilization for each device, so these were informed based on the expert opinion of our Egyptian collaborators (AAS, WA,SA). As Ain-Shams is a University Hospital where adherence to control measures is usually high, the probability of successful sterilization was set to 95% for all device types.

In addition to the baseline scenario exploring transmission within a high-resource setting, we explored the case of a hospital with lower resources and lower adherence to control measures. To that aim, we divided the quantity of available supply by two, and the resupply frequency was estimated with this new information. In addition, the probability of successful sterilization was reduced to 80% for all device types.

Our model was run over a year, representing 105,120 time steps. The main outcomes retrieved from our simulations were the daily HCV incidence rate as well as the yearly cumulative incidence at a hospital and ward level. In addition, we estimated the yearly portion of new HCV cases attributable to each type of device.

**2.6.2. Interventions: reinforced infection control.** We then simulated two different interventions based on HCV testing to reduce the risk of infection for hospitalized patients:

1. **A targeted ward-level systematic-screening intervention.** The three most at-risk wards (i.e., with the highest yearly estimated cumulative incidence) were identified after running baseline scenarios. Every patient entering one of these wards was assumed to be screened for HCV. Reinforced infection control was then implemented for identified positive patients, simulated as systematic successful sterilization of devices following use on these patients (i.e., the probability of successful sterilization was set to 1).

2. **A random screening upon admission intervention.** A random subset of patients entering the hospital were assumed to be screened for HCV. Reinforced infection control was then implemented for identified positive patients, simulated as systematic successful sterilization of devices following use on these patients.

The total number of tests performed in the three wards targeted by the systematic-screening intervention over a year was retrieved so that the number of random tests in the random screening upon admission intervention was set to be the same in order to compare these two scenarios. Test sensitivity and specificity were both assumed to be 100%.

**2.6.3. Sensitivity analysis.** Various sources of uncertainty might have interfered with the per-procedure risk estimations. In particular, the risk associated with surgery was estimated based on ORs reported in multiple studies. However, no information was available on whether this procedure described the procedure itself or took into account surgeries and other often associated sub-procedures (such as intubation or injection, for example) as a whole. Thus, a sensitivity analysis was performed in order to compare our baseline assumption (this risk is, in fact, associated with the act of surgery itself) with the case, considering that this risk already takes into account the risk of other associated sub-procedures, and then deleting in our data all procedures occurring in the operating room except the ones reported as surgeries.

**2.6.3. Extension to HBV.** To illustrate how the model may also be used to study HBV transmission in hospitals, we applied it to the case of an Ethiopian hospital, in which high levels of HBV prevalence were found recently. We assumed a similar hospital structure with three modified input parameters:

1. **The initial prevalence**. We retrieved age-specific HBV seroprevalence from Mohammed et al. (2022) [10], who explored HBV infections in two Ethiopian hospitals.

2. **The per-procedure infection risks**. We estimated per-procedure infection risk distributions for HBV by multiplying those computed for HCV by the ratio of the probability of HBV infection after exposure to contaminated blood (6–30%) over the same probability for HCV (1.8%) [11].

3. **The duration of the eclipse phase.** Its value for HBV was retrieved from the literature and estimated to be 15 days [12]

More information on these calculations is available as Supplementary Material. All other parameters were kept unchanged. We explored the same scenarios for HCV (high-resource vs. low-resource hospitals) and assessed the same interventions (systematic- vs. random screening).

**2.6.6. Model simulations.** This model was coded in C++ using the *Rcpp* interface in R version 4.3.1 [13,14]. All simulations were performed using the same R version. The number of simulations needed to catch most of the variability produced by the model was estimated by studying the convergence of the cumulative mean of the number of yearly cases for the low-resource setting scenario. S8 Fig shows that the mean converges after approximately 100 simulations. Therefore, each scenario was simulated 100 times. All simulations were distributed across 18 CPU threads (running at 4.8Ghz with 64Gb of RAM) using the R packages *'parallel'* [14] *and 'doParallel'* [15]. The average computation time for 100 simulations of a scenario was 25 minutes.

## 3. Results

### 3.1. HCV in Egypt

**3.1.1. Baseline scenarios.** The predicted yearly number of HCV acquisitions among hospitalized patients differed highly depending on the scenario.

In the high-resource setting case, this number was estimated to be around 1.1 cases per 100,000 patients/year (95% PI [0–4.0]) (Fig 2A). The yearly incidence rate was the highest in the OR (3.94 cases per 100,000 per year, 95% PI [0–17.0]), ER ICU (1.6 cases per 100,000 per year, 95% PI [0–1.6]), and ER (0.08 cases per 100,000 per year, 95% PI [0–1.6]) wards (Fig 2B), though accounting for 0.72, 0.08 and 0.04 annual cases on average, respectively. The type of device associated with the largest number of HCV contaminations was the endotracheal tube, with 90% of all cases on average (Fig 3A).

In the low-resource setting, the average yearly number of cases was estimated at around 60 per 100,000 patients/year (95% PI [46.25–87.01]) (Fig 2C). The ER ICU, OR, and ER wards were associated with the highest yearly incidence, with 196.7 cases per 100,000 patients/year (95% PI [0–521.72]), 64 cases per 100,000 patients/year (95% PI [19.9–109.6]) and 29.6 cases per 100,000 patients/year (95% PI [18.8–49.3]) respectively (Fig 2D). IV cannulas were associated with the highest number of HCV contaminations, representing around 22% of all HCV infections on average, followed by lancets (around 18% of all infection cases) (Fig 3B).

**3.1.2. Interventions.** In each case (i.e., high-resource or low-resource setting), the efficacy of the targeted ward-level systematic screening interventions was superior to that of the upon-admission random-testing interventions (Fig 4).

For both the high and low-resource setting cases, the OR, ER ICU, and ER wards were the most at-risk ones. A total average number of 53,640 patients (72%) went through these three wards during their hospitalization.

For the high-resource setting case, performing systematic screening in these three most at-risk wards reduced the annual number of HCV acquisition cases by 100% on average (95% PI [100–100]) compared to the baseline scenario, whereas random testing in 72% of patients upon admission reduced the annual number of cases by 81.3% (95% PI 0–100]).

Trends were similar for the low-resource setting case. Performing systematic screening in the three most at-risk wards allowed a reduction of 83.3% (95% PI [72.4–92.9]). Performing random screening in 72% of newly admitted patients reduced the annual number of cases by 73.2% (95% PI [57.1–85.1]).

**3.1.3. Sensitivity analysis.** When considering that our estimate of HCV infection risk associated with surgery already takes into account the risk of other associated sub-procedures, the yearly cumulative incidence was reduced by 85.6% in the high-resource setting baseline

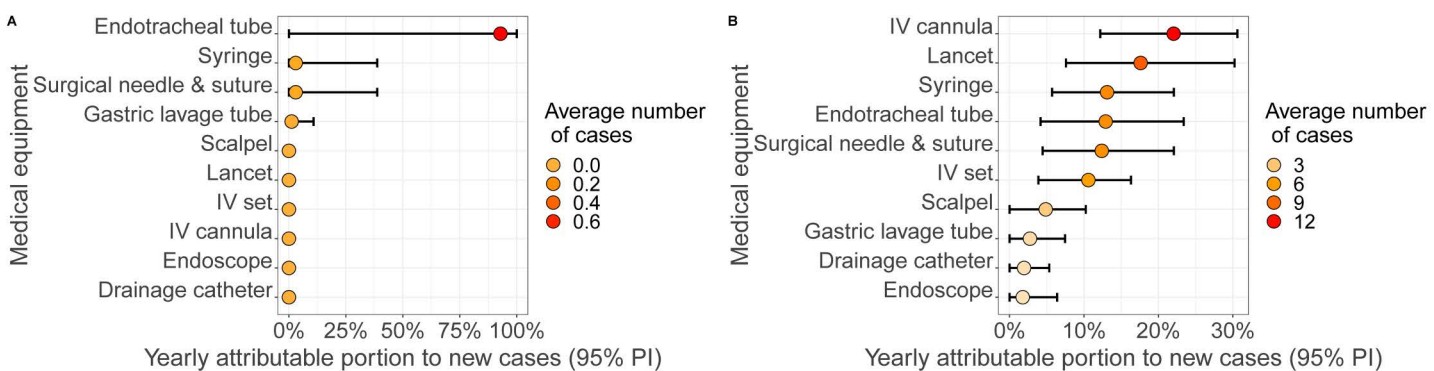

**Fig 2. Results of the model for baseline scenarios (HCV case).** (A) and (C): Daily incidence rate for the high- and low-resource hospitals, respectively. (B) and (D) Yearly cumulative incidence (mean and 95% PI) and average number of cases for each ward, ranked by mean cumulative incidence values.

**Fig 3. Yearly attributable portion to new cases for each device, in (A) Baseline scenario for the high-resource and (B) Baseline scenario for the low-resource hospital.**

scenario (with 0.16 cases per 100,000 patients/year; 95% PI [0–1.88]) and there was no change in the low-resource setting baseline scenario (60.3 cases per 100,000 patient/year; 95% PI [42.6–85.4]), compared to the baseline case where each of the procedures occurring in the operating room was taken into account individually.

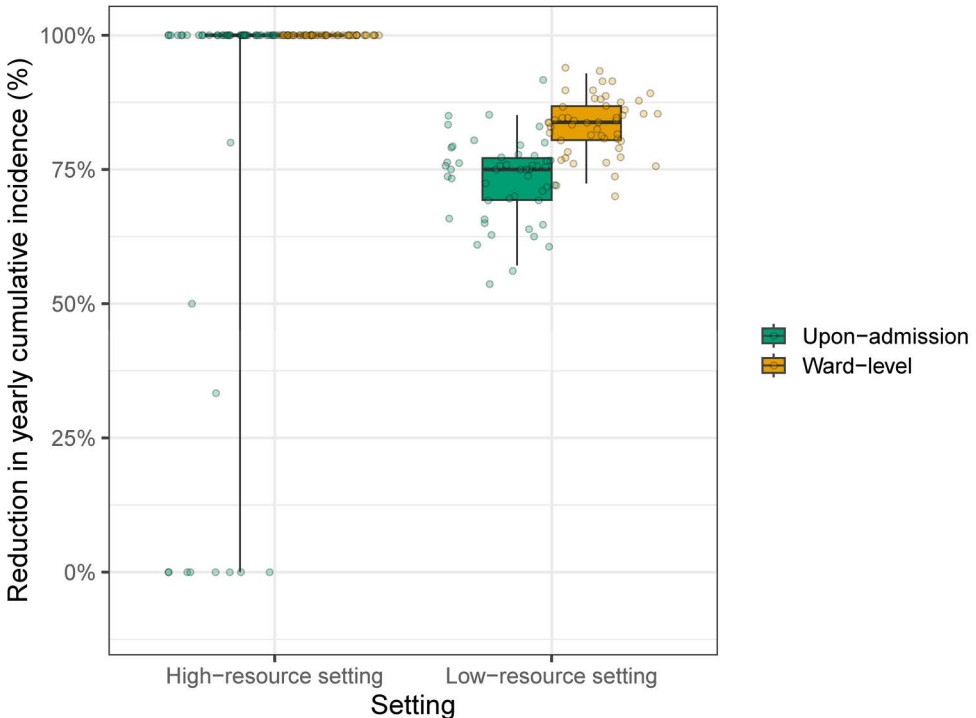

**Fig 4. Reduction in the predicted yearly cumulative incidence, by comparison with the baseline scenarios, obtained by two different intervention strategies implemented in (A) high-resource and (B) low-resource hospitals.** Baseline scenarios correspond to the no-intervention scenarios. For both high-resource and low-resource settings, on average, 53,640 patients (72%) were screened (i) either systematically (in the three most at-risk wards) or (ii) randomly upon admission. Median relative reductions computed over 100 simulations, along with the first and third quantiles (boxes) of the associated distributions and the 95% prediction intervals, are depicted.

## 3.2. HBV in Ethiopia

**3.2.1. Baseline scenarios.** The predicted number of HBV acquisitions among hospitalized patients again differed highly between low- and high-resource hospital settings (S5 Fig).

In high-resource settings, this number was estimated at 12.6 cases per 100,000 patients/year (95% PI [5.5–20.3]), with the OR ward being the most at-risk, with a yearly cumulative incidence of 55.6 cases per 100,000 patients/year (95% PI [18.3–92.4]), followed by the ER ICU ward (7.7 cases per 100,000 patient/year, 95% PI [0–77.7]) and the Neurosurgery ward (3.4 cases per 100,000 patient/year, 95% PI [0–42.3]). The type of device associated with the highest number of HCV contaminations was the endotracheal tube, with more than 85% of all cases on average (S6 Fig).

For low-resource settings, the yearly cumulative incidence reached 677.5 cases per 100,000 patient/year (95% PI [584.5–749.6]), with the ER ICU ward again being one of the most at-risk with a yearly cumulative incidence of 2,243.7 cases per 100,000 patient/year (95% PI [1,002.3–3,237.43]), followed by the OR ward (686.4 cases per 100,000 patient/year, 95% PI [530.1–850.8]), and the ER ward (344.4 cases per 100,000 patient/year, 95% PI [285.2–409.0]). IV cannulas were associated with the highest number of contaminations, accounting for around 22% of all contaminations, on average (S6 Fig).

**3.2.2. Interventions.** The same trends as for the HCV case were observed in the HBV case (S7 Fig). Systematic screening in the most at-risk wards (representing 72% of all patients) was found to reduce the number of cases by 97.1% (95% PI [83.0–99.3]) in the high-resource

hospital. On the other hand, the reduction was 71.3% on average (95% PI [37.5–98.0]) when randomly screening patients upon admission.

For the low-resource hospital, the reductions obtained by these same interventions targeting 72% of patients were 81.5% (95% PI [77.2–85.1]) and 71.1% (95% PI [64.4–78.3]), for the systematic and random screening interventions, respectively.

## 4. Discussion

Herein, we describe the framework of a novel agent-based model of the nosocomial transmission of bloodborne pathogens. Using data on hospital structure, within-hospital patient movements, and procedures undergone by patients, we were able to simulate the dynamics of patients and medical devices within the hospital. Applying the model to the Egyptian context suggested that the risk of HCV infection is low in a hospital with high resources and strong disinfection practices (less than 5 cases per 100,000 patients per year) but could be higher for a low-resource hospital, with up to approximately 6 cases per 10,000 patients per year. In a second illustrative case study considering HBV transmission in an Ethiopian hospital, we found similar trends but with higher absolute risk values.

Until now, and to our knowledge, no such model has been available in the literature, and all the previous modelling approaches to exploring bloodborne pathogen spread in healthcare settings were based on quantitative risk assessment [16,17]. However, dynamic models have already been used to study the transmission dynamics of HCV and other bloodborne pathogens in the community, especially within IDU networks, as reviewed by Cousien et al. (2015) [6]. Most of these models were fully compartmental, and only a few used an individual-based approach.

Our results suggest that hospitals might still generate new HCV and HBV contaminations, especially in low resource settings. This is in line with recent studies highlighting a persisting increased risk of HBV and HCV infection associated with medical and surgical procedures [2].

On the one hand, the incidence of infection observed in each ward depends on several factors, such as the prevalence on admission to these wards, the number of devices available (which determines how these devices are reused), the average number of procedures undergone by patients, and the specific risk of transmission during each of these procedures. In our case, it is not entirely surprising to find that the emergency ICU is one of the wards with the highest risk, as this risk seems to be driven by the high number of procedures performed on patients in this ward (around 33 procedures per patient, which is making it the ward with the highest number of procedures performed on patients). In addition, the risk of infection in the OR ward appears to be the result of the variety of procedures carried out there, for which there is often a lack of medical supplies.

On the other hand, the ranking of the most at-risk medical devices is strongly driven by the quantity of available devices and their frequency of use within the hospital. Finding IV cannulas and lancets to be the most at-risk devices in the low-resource setting scenario is not very surprising as they are used in 19 and 23 different wards, and they are lacking in 68% and 40% of these wards, respectively. In addition, the fact that we find found some devices to be more at-risk for transmission of HCV or HBV in the lower-resource setting scenario is due to the overall lower number of devices available in this scenario. Some devices ended up requiring reprocessing and reuse in the lower-resource setting scenario, but not in the higher-resource setting scenario. Community incidence of HCV in rural villages in Egypt was estimated at 37/100,000 per year in 2018, soon after implementing a mass HCV screening campaign. Even if this may not be fully comparable with our results, our estimations remain in the same order of magnitude [18]. We showed that nosocomial acquisitions could, however,

be tackled by improving control measures and better allocating financial resources to make more sterile devices available for hospitals.

Some limitations related to the model structure and data used to inform the model may be highlighted.

First, the model only accounts for between-patient transmission without considering transmission from healthcare workers to patients or from patients to healthcare workers. Contamination through these routes has been reported, leading us to potentially underestimate acquisition risks. However, such cases are usually quite rare [19].

In addition, our results highly depend on the assumed per-procedure risks of transmission through blood contact. For the HCV case study, these risks were informed by a previously published meta-analysis [4]. However, as mentioned in the methods section, they might still have been over or underestimated. In particular, our sensitivity analysis showed that the estimated yearly cumulative incidence could vary depending on the assumption made on the risk associated with surgery: considering that this risk takes into account the risk of other associated sub-procedures led to estimating a lower yearly cumulative HCV incidence. On a related note, we assumed a similar risk of bloodborne pathogen transmission for endotracheal intubation and endoscopy, possibly leading us to overestimate the risk associated with endotracheal tubes. In fact, although blood contact may occur during this procedure (e.g., caused by oral or tracheal trauma), it could be minimal and we found no evidence in the literature of bloodborne infections following endotracheal intubation. To assess the effect of a lower assumed risk associated with this procedure on the results of our model, we examined reductions of 50% and 90% of the baseline risk. Overall, risk reductions led to slightly lower yearly cumulative incidences in both high- and low-resource settings (S11 Fig). In both reduction scenarios, in the high-resource setting, this had no effect on the ranking of the most at-risk devices, as the available quantity of endotracheal tubes was still largely insufficient compared to other devices (S12 Fig); the ward ranking also remained unchanged (S13 Fig). In the low-resource setting, endotracheal tubes became ranked as the 6th or 9th most at risk devices (compared to 4th in the baseline scenario) when the assumed per-procedure risk decreased by 50 or 90% (S12 Fig). Finally, the OR ward (in which most endotracheal intubations are performed) was estimated to be the third most at-risk ward rather than the second one in low-resource settings, with average reductions in yearly cumulative incidence of 31.9% and 59.4%, respectively (S13 Fig)."

Our model allowed us to assess the effectiveness of ward-level systematic and upon-admission random screening in the hospital. Systematic screening targeted at the most at-risk wards was overall more efficient in reducing the yearly number of cases. Nevertheless, we assumed perfect disinfection of devices used on positive patients, which might be difficult to reach in practice. Other interventions could be assessed using our model. In particular, a more realistic intervention could be device reallocation between wards, which could reduce the overall number of infections by setting the risk to zero in the most at-risk wards while potentially increasing it in other (e.g., with lower HCV prevalence) wards.

Then, the data used to inform the quantity of available devices within wards was approximated using the available devices at the entire Ain Shams hospital level, assuming that these devices were allocated to wards proportionally to admissions. This may not reflect the real supply allocation. In addition, we did not have any data to inform device availability in low-resource hospitals, leading us to arbitrarily assume that the available quantity of devices in these low-resource hospitals was twice as low as for high-resource hospitals. Moreover, we did not have access to data for every type of medical devices. In particular, high-risk devices such as intraosseous catheters or bone marrow aspiration needles were not taken into account. Finally, probabilities of successful disinfection were chosen based on expert opinion rather

than relying on data collected within hospitals. However, the probability of efficient disinfection of 80% considered for endotracheal tubes in the low-resource setting scenario seems to be in line with a study by Elisa *et al.* [20] showing disinfection efficiencies of 71% and 83%. All these limitations stem from a lack of data to correctly inform the corresponding model parameters, forcing us to make assumptions. However, our aim here was mainly to describe a new modelling framework and to illustrate its potential applications rather than to inform public health decision-makers directly regarding HCV or HBV control in hospitals.

On a related note, our results may not accurately describe the current Egyptian situation as we used data from 2017. The current upon-admission prevalence of HCV-infected patients might be much lower, as a massive test and treatment program launched by the Egyptian government in 2018 led to the treatment of more than 2 million Egyptians, and the national prevalence in 2023 is expected to fall around 0.5% [21]. Nevertheless, many countries are still affected by HCV and other bloodborne pathogens, and the model we propose would be helpful to predict their burden within hospitals.

The advantage of this model lies in its high flexibility. It can account for heterogeneity between patient profiles by informing different transition matrices and could be applied to other hospitals in other contexts beyond those described here. In addition, computation times remain acceptable with respect to the model complexity.

Some countries still report a high HCV or HBV prevalence. A recent meta-analysis focusing on HBV prevalence in sub-continental countries showed that Pakistan and India still have a national prevalence of over 5% [22]. Several of the devices mentioned in our results are intended to be only used once [23,24], but compliance with these recommendations is still not perfect, with contamination events still being reported, even in high-income countries [25,26]. In addition, outbreaks of infections still occur nowadays, even in developed countries. Between 2006 and 2020, 91 outbreaks of transmission of HBV and HCV within hospitals have been reported in Europe, corresponding to 442 cases of infections [27]. In the USA, such events were reported 66 times between 2008 and 2019, corresponding to more than 500 new infections [28]. Due to its flexibility, our model could simulate bloodborne pathogen transmission within a wide array of settings and thus help predict such outbreaks, track past contamination events, and assess the effectiveness of intervention measures.

To conclude, the modelling tool we propose may help study the spread of bloodborne pathogens at a hospital level and assess the efficacy of multiple intervention measures in reducing their transmission, as well as the associated costs. It could help implement more efficient prevention measures in hospitals and, in turn, could contribute to achieving WHO HCV elimination targets, especially in Egypt where community transmission has been largely controlled. Our model could help study the transmission of other bloodborne pathogens, such as Zika virus or prions, which are poorly understood. In addition, as experienced with the COVID-19 pandemic, emerging pathogens can lead to a substantial public health burden. Being able to model the transmission of potential emerging bloodborne pathogens could help predict the associated disease burden in hospitals and help implement efficient public health policies.

## Supporting information

**S1 Table. Ward IDs and names.**
(DOCX)

**S2 Table. Matrix of association between devices and procedures.** For each device (in lines), each cell indicates whether it is used for each procedure (in columns).
(DOCX)

**S3 Table. Transition matrix for patients hospitalized in the surgery department.** Each cell provides the proportion of patients moving from ward i (in line) to ward j (in column) at each time-step.
(DOCX)

**S4 Table. Transition matrix for patients hospitalized in the internal medicine department.** Each cell provides the proportion of patients moving from ward i (in line) to ward j (in column) at each time-step.
(DOCX)

**S5 Table. Ward-specific probabilities of undergoing each of the procedures in the surgery department.** Each cell provides the probability of undergoing procedure j (in column) in ward I (in line) at each time-step.
(DOCX)

**S6 Table. Ward-specific probabilities of undergoing each of the procedures in the internal medicine department.** Each cell provides the probability of undergoing procedure j (in column) in ward I (in line) at each time-step.
(DOCX)

**S7 Table. Yearly initial quantity of new devices in each ward for the high-resource setting.**
(DOCX)

**S8 Table. Yearly initial quantity of previously used devices in each ward for the high-resource setting.**
(DOCX)

**S9 Table. Yearly initial quantity of new devices in each ward for the low-resource setting.**
(DOCX)

**S10 Table. Yearly initial quantity of previously used devices in each ward for the low-resource setting.**
(DOCX)

**S1 Fig. Ratio of the total quantity of available devices over the necessary quantity required by procedures, for each device type in each ward in the high-resource hospital.** There are insufficient devices to cover the hospital's needs when the value <1.
(PNG)

**S2 Fig. Ratio of the total quantity of available devices over the necessary quantity required by procedures, for each device type in each ward in the low-resource hospital.** There are insufficient devices to cover the hospital's needs when the value <1.
(PNG)

**S3 Fig. Yearly device use (columns) in each ward (rows), based on observed procedures.**
(PNG)

**S4 Fig. Distribution of the risk of HCV infection associated with different groups of procedures considered in our work, based on a previously published meta-analysis (Henriot *et al.*, 2022)** [4]
(PNG)

**S5 Fig. Model predictions for baseline scenarios (HBV case study).** (A) and (B): Daily incidence rate for the high- and low-resource hospital, respectively. (C) and (D) Yearly cumulative incidence (mean and 95% PI) and average number of cases for each ward, ranked by mean cumulative incidence values.
(PNG)

**S6 Fig. Yearly portion of new cases attributable to each device, in (A) Baseline scenario for the high-resource and (B) Baseline scenario for the low-resource hospital.**
(PNG)

**S7 Fig. Reduction in cumulative incidence for two different intervention strategies for (A) High-resource and (B) Low-resource hospitals.** Baseline scenarios correspond to the no-intervention scenarios. In both high-resource and low-resource settings, 53,640 patients (72%) are screened (i) either systematically (in the three most at-risk wards) or (ii) randomly upon admission.
(PNG)

**S8 Fig. Cumulative mean number of predicted annual cases over 1 to 200 simulations.** This graph shows that the mean seems to converge after 100 simulations.
(PNG)

**S9 Fig. Empirical (red) and estimated (blue) distributions of the number of procedures per patient.** The average number of procedures extracted from the data was of 11.22 (95% CI [9.90–12.54]; median: 7) and was estimated at 11.36 (95% CI [11.26–11.46]; median: 7) using our model.
(PNG)

**S10 Fig. Empirical (red) and estimated (blue) distributions of length of stay (in days).** The average number of procedures extracted from the data was of 4.60 (95% CI [4.24–4.97]; median: 3.05) and was estimated at 4.68 (95% CI [4.65–4.72]; median: 3) using our model.
(PNG)

**S11 Fig. Daily incidence rates when the risk associated with endotracheal intubation is reduced by 50% and 90% in the high-resource setting case (a and b), and in the low-resource setting case (c and d).**
(PNG)

**S12 Fig. Yearly cumulative incidence (mean and 95% PI) and average number of cases for each ward, ranked by mean cumulative incidence values when reducing the risk associated with endotracheal intubation by 50% and 90% in the high-resource setting case (a and b), and in the low-resource setting case (c and d).**
(PNG)

**S13 Fig. Yearly portion of new cases attributable to each device, when reducing the risk associated with endotracheal intubation by 50% and 90% in the high-resource setting case (a and b), and in the low-resource setting case (c and d).**
(PNG)

**S1 Text. Method used to estimate the resupply frequency**
(DOCX)

## Author contributions

**Conceptualization:** Paul Henriot, Kévin Jean, Laura Temime.

**Data curation:** Wagida Anwar, Samia A. Girgis, Maha El Gaafary.

**Formal analysis:** Paul Henriot, Laura Temime.

**Funding acquisition:** Paul Henriot.

**Methodology:** Paul Henriot, Kévin Jean, Laura Temime.

**Software:** Paul Henriot.

**Supervision:** Kévin Jean, Laura Temime.

**Validation:** Wagida Anwar, Maha El Gaafary, Kévin Jean, Laura Temime.

**Visualization:** Paul Henriot, Kévin Jean, Laura Temime.

**Writing – original draft:** Paul Henriot, Kévin Jean, Laura Temime.

**Writing – review & editing:** Paul Henriot, Mohamed El-Kassas, Wagida Anwar, Samia A. Girgis, Maha El Gaafary, Kévin Jean, Laura Temime.

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
