## [Decision Letter · Decision Letter 0]

9 Jul 2024

Dear Mr Henriot,

Thank you very much for submitting your manuscript "An agent-based model to simulate the transmission dynamics of bloodborne pathogens within hospitals" for consideration at PLOS Computational Biology.

As with all papers reviewed by the journal, your manuscript was reviewed by members of the editorial board and by several independent reviewers. In light of the reviews (below this email), we would like to invite the resubmission of a significantly-revised version that takes into account the reviewers' comments.

When considering your revision, please make sure to address the clinical credibility of your findings, as raised by Reviewer 1, as well as the more pragmatic, modeling-focused comments from Reviewer 2.

We cannot make any decision about publication until we have seen the revised manuscript and your response to the reviewers' comments. Your revised manuscript is also likely to be sent to reviewers for further evaluation.

Sincerely,

Eric Lofgren, MSPH, PhD

Academic Editor

PLOS Computational Biology

Thomas Leitner

Section Editor

PLOS Computational Biology

When considering your revision, please make sure to address the clinical credibility of your findings, as raised by Reviewer 1, as well as the more pragmatic, modeling-focused comments from Reviewer 2.

Reviewer's Responses to Questions

**Comments to the Authors:**

Reviewer #1: The review is uploaded as an attachment

Reviewer #2: Henriot et al have developed an agent-based model of the transmission of blood-borne pathogens in hospital wards. This model includes a detailed treatment of device use, contamination, and disinfection, and movement between hospital wards based on data collected from a hospital in Egypt. The authors then use this model to estimate incidence in different settings, and to evaluate the effect of screening interventions, finding that incidence could potentially be quite high in low-resource settings but that entry- and ward-level screening could lead a large reduction in incidence.

The strengths of this paper include the clear writing, visually appealing and easily understood figures, and the novel approach to an under researched area. The model itself is carefully designed and is explained well in the manuscript. Although it is quite complex, the complexity is justified and its parameters are largely well-informed by the data. There are no major weaknesses, but some minor weaknesses / issues include the lack of availability of the code and the lack of validation of model results with real data. I also think that the intuition behind the results could be explained a little more carefully for readers not familiar with this area (e.g. why the model predicts that some procedures / tools carry higher risk, and whether this is unexpected or not).

Please see below for some more detailed suggestions.

Introduction

• L53: please could you remove the sentence beginning “To our knowledge”, here and in other places, as it is impossible to verify these claims and they are unnecessary anyway.

Methods

• In table 1B, please state the units of the eclipse phase duration

• L129: is there any chance that the alpha_ip will ever sum to greater than 1? e,g, if patients undergo multiple procedures

• L186: I’m not sure I understand this equation.

◦ Does it make sense to sum the probabilities of devices in the summation? Isn’t there a risk that P_{S→E} is greater than 1, e.g. if lots of contaminated devices are present in the ward, the probability of infection given contamination is high, or lots of devices are required in the procedure?

◦ Why isn’t the expression something like:

1 – product(1 – r_p * P(X^k))? I appreciate that this expression will be similar to yours for small probabilities, but that the probabilities are small is not clear to the reader at this point. So it might be worth clarifying in the text that the expression is an approximation and is appropriate because these probabilities are small. Unless I am misunderstanding something.

• I think the authors should endeavour to make the code available on GitHub, with synthetic data as the raw data cannot be shared.

Results

• What is the intuition / reasoning behind which wards and which procedures caused the highest numbers of infections (Fig 2)?

• Why does the distribution of which tools have the biggest impact on incidence change between resource settings (Fig 3)?

• I know it’s only two data points, but it seems perhaps misleading that for some simulations in high-resource settings, the intervention increases incidence, which seems implausible for this type of intervention. I presume that it’s because of differing stochastic events between paired simulations or something like that, but it makes me wonder if the results make the effect of the intervention appear more uncertain that they should. I wonder if there was a better way to pair simulations (e.g. using random seed, or ranking them by incidence), or perhaps to take an average across a (greater) number of simulations before comparing? Just some thoughts.

• Are there any data available on hospital incidence of HCV or HBV with which to (qualitatively) validate your results? It’s hard to know (for me at least) if what we are seeing in Figure 1 is plausible or not.

Discussion

• L520: While I am sure this model could be applied, I’m not sure it is easy given the data requirements and complexity of the model – perhaps just remove the word ‘easily’

• L521: Could you report computation times somewhere in the manuscript? (Apologies if I missed it)

• L536: I would reword this sentence – making WHO elimination targets more easily achieved is not a prevention measure, and anyway it’s not clear that this type of model could help achieving elimination targets without establishing the overall importance of hospital transmission compared to community transmission.

**Have the authors made all data and (if applicable) computational code underlying the findings in their manuscript fully available?**

Reviewer #1: None

Reviewer #2: **No: ** They state that data is not shareable. However, I think the code could still be shared, preferably with synthetic data. I have requested this in my comments.

PLOS authors have the option to publish the peer review history of their article (what does this mean? ). If published, this will include your full peer review and any attached files.

**Do you want your identity to be public for this peer review?** For information about this choice, including consent withdrawal, please see our Privacy Policy .

Reviewer #1: **Yes: ** Herberth Maldonado Briones

Reviewer #2: No
---

## [Decision Letter · Decision Letter 1]

15 Nov 2024

PCOMPBIOL-D-24-00422R1An agent-based model to simulate the transmission dynamics of bloodborne pathogens within hospitalsPLOS Computational Biology Dear Dr. Henriot, Thank you for submitting your manuscript to PLOS Computational Biology. After careful consideration, we feel that it has merit but does not fully meet PLOS Computational Biology's publication criteria as it currently stands. Therefore, we invite you to submit a revised version of the manuscript that addresses the points raised during the review process. Please consider Reviewer 1's lingering comments and address them. Please submit your revised manuscript within 30 days Jan 15 2025 11:59PM. If you will need more time than this to complete your revisions, please reply to this message or contact the journal office at ploscompbiol@plos.org. Please include the following items when submitting your revised manuscript:* A rebuttal letter that responds to each point raised by the editor and reviewer(s). You should upload this letter as a separate file labeled 'Response to Reviewers'. This file does not need to include responses to formatting updates and technical items listed in the 'Journal Requirements' section below.* A marked-up copy of your manuscript that highlights changes made to the original version. You should upload this as a separate file labeled 'Revised Manuscript with Track Changes'.* An unmarked version of your revised paper without tracked changes. You should upload this as a separate file labeled 'Manuscript'. If you would like to make changes to your financial disclosure, competing interests statement, or data availability statement, please make these updates within the submission form at the time of resubmission. Guidelines for resubmitting your figure files are available below the reviewer comments at the end of this letter. We look forward to receiving your revised manuscript. Kind regards, Eric Lofgren, MSPH, PhDAcademic EditorPLOS Computational Biology Thomas LeitnerSection EditorPLOS Computational Biology

Feilim Mac Gabhann

Editor-in-Chief

PLOS Computational Biology

Jason Papin

Editor-in-Chief

PLOS Computational Biology

 **Journal Requirements:** **Additional Editor Comments (if provided):****Reviewers' comments:** Reviewer's Responses to Questions

**Comments to the Authors:**

Reviewer #1: Agreed with your review. I am glad that this process helped to clarify your results.

Just a last observation: you argue based on expert opinion that the risk of bloodborne pathogen transmission is similar for endoscopy and endotracheal intubation, but I find it weak:

Endoscopy procedure was identified as a low risk of HCV transmission 1.48(0.95−2.3) in previous studies. Endoscopy is an ivasive procedure that uses a rigid or flexible probe and these procedures frequently are related with gastric erosions, ulcers, and/or bleeding. These probes are have holes and channels and since always are reused needs standardized protocols for cleaning, disinfection and chemical sterilization. Several reports of device-associated infection due to endoscopes are described in the literature.

In contrast, endotraqueal intubation is an invasive procedure that uses a laryngoscope, and insertion of cuffed or non-cuffed endotracheal tubes. These devices has two or three holes and the contact with blood is minimal and can be related to oral trauma, mucosal tracheal erosions or infrequently by pulmonary bleeding. Again, no device-associated infections due to reprocessed endotracheal tubes are described or reported.

Please justify your expert opinion based on the available evidence and rationale.

Otherwise I suggest to adjust the model, e.g. adding the probability of bleeding while endotracheal tube is in place, time of exposure (in days where greater risk comes with more exposure).

Reviewer #2: The authors have adequately addressed all of my initial minor concerns, and I have no further comments.

**Have the authors made all data and (if applicable) computational code underlying the findings in their manuscript fully available?**

Reviewer #1: Yes

Reviewer #2: Yes

PLOS authors have the option to publish the peer review history of their article (what does this mean? ). If published, this will include your full peer review and any attached files.

**Do you want your identity to be public for this peer review?** For information about this choice, including consent withdrawal, please see our Privacy Policy .

Reviewer #1: **Yes: ** Herberth Maldonado

Reviewer #2: **Yes: ** Sean Cavany

---

## [Editor Report · Decision Letter 2]

5 Feb 2025

Dear Mr Henriot,

We are pleased to inform you that your manuscript 'An agent-based model to simulate the transmission dynamics of bloodborne pathogens within hospitals' has been provisionally accepted for publication in PLOS Computational Biology.

Best regards,

Eric Lofgren, MSPH, PhD

Academic Editor

PLOS Computational Biology

Thomas Leitner

Section Editor

PLOS Computational Biology

---

## [Editor Report · Acceptance letter]

PCOMPBIOL-D-24-00422R2

An agent-based model to simulate the transmission dynamics of bloodborne pathogens within hospitals

Dear Dr Henriot,

I am pleased to inform you that your manuscript has been formally accepted for publication in PLOS Computational Biology. Your manuscript is now with our production department and you will be notified of the publication date in due course.

With kind regards,

Anita Estes
